# Modern Aspects of Immunotherapy with Checkpoint Inhibitors in Melanoma

**DOI:** 10.3390/ijms21072367

**Published:** 2020-03-30

**Authors:** Vera Petrova, Ihor Arkhypov, Rebekka Weber, Christopher Groth, Peter Altevogt, Jochen Utikal, Viktor Umansky

**Affiliations:** 1Clinical Cooperation Unit Dermato-Oncology, German Cancer Research Center (DKFZ), 69120 Heidelberg, Germany; Vera.Petrova@medma.uni-heidelberg.de (V.P.); Ihor.Arkhypov@medma.uni-heidelberg.de (I.A.); rebekka.weber@dkfz.de (R.W.); c.groth@dkfz.de (C.G.); p.altevogt@dkfz.de (P.A.); j.utikal@dkfz.de (J.U.); 2Department of Dermatology, Venereology and Allergology, University Medical Center Mannheim, University of Heidelberg, 68167 Mannheim, Germany

**Keywords:** melanoma, immunotherapy, immune checkpoint inhibitors, immunosuppression, tumor microenvironment

## Abstract

Although melanoma is one of the most immunogenic tumors, it has an ability to evade anti-tumor immune responses by exploiting tolerance mechanisms, including negative immune checkpoint molecules. The most extensively studied checkpoints represent cytotoxic T lymphocyte-associated protein-4 (CTLA-4) and programmed cell death protein 1 (PD-1). Immune checkpoint inhibitors (ICI), which were broadly applied for melanoma treatment in the past decade, can unleash anti-tumor immune responses and result in melanoma regression. Patients responding to the ICI treatment showed long-lasting remission or disease control status. However, a large group of patients failed to respond to this therapy, indicating the development of resistance mechanisms. Among them are intrinsic tumor properties, the dysfunction of effector cells, and the generation of immunosuppressive tumor microenvironment (TME). This review discusses achievements of ICI treatment in melanoma, reasons for its failure, and promising approaches for overcoming the resistance. These methods include combinations of different ICI with each other, strategies for neutralizing the immunosuppressive TME and combining ICI with other anti-cancer therapies such as radiation, oncolytic viral, or targeted therapy. New therapeutic approaches targeting other immune checkpoint molecules are also discussed.

## 1. Introduction

The concept of cancer immunosurveillance is based on the fact that tumor cells can be recognized and eliminated by immune system [1,2]. Immunogenicity of malignant melanoma is based on a high ultraviolet-driven mutational burden [3]. This leads to the overexpression of tumor specific antigens enabling the formation of the antigen specific immune response [4,5]. However, development of aggressive metastatic melanoma shows that tumors are edited by the immune system, and selected resistant variants could escape the immune control [6,7]. Therefore, several immune-based therapeutic approaches such as vaccination [8], adoptive transfers [9] and immune checkpoint-blockade [10] were applied, aiming at reinvigorating anti-tumor immune response and improving survival of advanced-stage melanoma patients [11].

The most studied negative immune checkpoint molecules and broadly accepted targets for immunotherapy are cytotoxic T lymphocyte-associated protein-4 (CTLA-4) and programmed cell death protein 1 (PD-1). CTLA-4 is upregulated on the T cell surface early during activation in lymph nodes, binds to CD80/CD86 reducing co-stimulation through CD28 and functions as a negative downstream loop for T cell receptor (TCR) signaling [12]. PD-1 interaction with its ligands PD-L1 and PD-L2 inhibits effector T cell functions in peripheral tissues [13]. Playing a pivotal role in the maintenance of self-tolerance under physiological conditions, these checkpoint molecules could be exploited by tumors to evade the immune responses. Hence, inhibiting such interactions could reactivate anti-tumor immune reactions [14]. Moreover, the combination of anti-CTLA-4 and anti-PD-1 antibodies was shown to work synergistically by expanding activated effector CD8 T cells [15,16]. Another approach was shown to implicate the combination of PD-L1-CD80 heterodimerization and the suppression of the CTLA-4/CD80 axis [17]. Currently used antibodies to target CTLA-4 are ipilimumab and tremelimumab, to target PD-1 are nivolumab, pembrolizumab, cemiplimab and to target PD-L1 are atezolizumab and avelumab [14,18,19].

This review will focus on current achievements in the therapy with immune checkpoint inhibitors (ICI) in melanoma and will discuss the strategies to improve of treatment efficacy by combining ICI with other therapies.

## 2. Therapeutic Effects of Immune Checkpoint Inhibitors

Latest clinical guidelines on melanoma management consider immune checkpoint blockade (anti-PD-1 alone or in combination with anti-CTLA4) as a first-line treatment option for unresectable stage III and IV melanoma patients [20,21]. In cases of resectable melanoma, anti-PD1 agents are prescribed as well in an adjuvant setting [22]. This treatment is currently investigating in a neoadjuvant setting [23].

Since the responses of tumors to immunotherapy and chemotherapy are different, immune-related response criteria and immune-response evaluation criteria in solid tumors were developed [24,25]. Such criteria improve the evaluation of additional response patterns during immunotherapy such as pseudoprogression. Currently achieved response to ICI treatment of melanoma patients reached 52% for pembrolizumab and 58% for combination of nivolumab and ipilimumab [26,27,28]. The 5-year survival rate was reported to be 41% and 52% in these two trials, respectively. These therapeutic achievements were associated with a high toxicity up to 59% of grade 3 and 4 adverse events in patients treated with the combination of nivolumab and ipilimumab [27]. Another trial studied a ipilimumab combination with pembrolizumab, which does not yet belong to the approved settings. The objective response was achieved by 61% of patients, 1-year overall survival (OS) was 89%, and 1-year progression-free survival was 69%. Grade 3 and 4 adverse events occurred in 27% of patients [29]. These data represent a favorable effect of such combinations with increased response values and less high-grade adverse effects.

However, many patients remained resistant to ICI therapy since tumor cells could develop resistance to anti-tumor immune reactions or induce a profound immunosuppression in the tumor microenvironment [30].

## 3. Tumor Cells Evade Immune Responses

A characteristic gene profile was described for melanoma cells resistant to ICI. It includes the repression of genes, which control antigen presentation and interferon (IFN)-γ signaling as well as the induction of genes regulating epithelial-mesenchymal transition, remodeling of extracellular matrix, cell adhesion and angiogenesis [31,32,33,34]. Interestingly, down-regulation of major histocompatibility complex (MHC) class I protein expression was found to be associated with the resistance to anti-CTLA-4, but not to anti-PD-1 therapy [35]. In the same work, MHC class II expression in >1% melanoma cells was shown to predict response to anti-PD-1, but not to anti-CTLA-4 therapy. This suggests that tumor cells disrupt antigen presentation limiting the efficient anti-tumor response. In fact, anti-PD-1 blockade before antigen priming of T cells leads to accumulation of the dysfunctional PD-1^+^CD38^hi^CD8^+^ cells abolishing the effects of the therapy [36]. Moreover, tumor cells can prevent the formation of anti-tumor T cell memory in the draining lymph node by secreting PD-L1-bearing extracellular vesicles (EV), contributing to the resistance to anti-PD-1 antibodies [37].

## 4. Immunosuppressive Tumor Microenvironment as an Important Factor of ICI Treatment Failure

A deeper investigation of the immunosuppressive networks within the TME could help to understand the limitations of ICI treatment and to develop strategies for increasing treatment efficiency. Immunosuppression in the TME is mediated by various cells and soluble factors described below.

### 4.1. Myeloid-Derived Suppressor Cells (MDSC)

MDSC represent a heterogeneous population of immunosuppressive myeloid cells, generating under chronic inflammation conditions and cancer and accumulating in the TME [38]. In humans, three MDSC subsets have been described: CD11b^+^CD14^+^HLA-DR^low/−^CD15^−^Lin^−^ monocytic (M-MDSC), CD14^−^CD11b^+^CD15^+^HLA-DR^low/−^Lin^−^ polymorphonuclear (PMN) MDSC, and HLA-DR ^low/−^CD33^dim^CD66b^−^Lin^−^ early-stage MDSC (e-MDSC) [39]. MDSC could inhibit anti-tumor functions of T and natural killer (NK) cells via different mechanisms. They can express PD-L1 and FasL and cause T cell anergy and apoptosis [40]. The induction of hypoxia-inducible factor-1α (HIF-1α) through transforming growth factor-β (TGF-β) and hypoxic conditions leads to the upregulation of the ectoenzymes CD39 and CD73, producing immunosuppressive adenosine in the extracellular space [40,41]. Reactive oxygen species (ROS) and nitric oxide (NO) produced by MDSC induce T cell apoptosis and the down-regulation of TCR ζ-chain expression [41,42]. Furthermore, MDSC can stimulate regulatory T cell (Treg) activity [43].

Previous studies demonstrated that high frequency of MDSC in the peripheral blood of advanced melanoma patients correlated with disease progression, decreased overall and progression free survival as well as decreased efficacy of immunotherapy, making them a promising therapeutic target [44,45,46,47]. There are different ways to suppress the immunosuppressive activity of MDSC [48]. Normalization of myelopoiesis and depletion of immunosuppressive MDSC could be achieved by using all-trans retinoic acid (ATRA) [49,50], tyrosine-kinase inhibitors [51,52] or some chemotherapeutic agents such as gemcitabine or paclitaxel [53,54].

Another approach of targeting MDSC represents an inhibition of their immunosuppressive activity. Based on the preclinical data showing that phosphodiesterase (PDE)-5 inhibitor sildenafil could suppress MDSC activity, enhance T cell functionality and prolong survival of melanoma-bearing mice [55,56], another PDE-5 inhibitor tadalafil was applied in advanced, therapy-resistant melanoma patients. Therapy was well-tolerated, and 25% of treated patients showed stable disease (SD) with the progression free survival (PFS) of 4.6 months [57]. Moreover, patients with SD showed increased infiltration of activated CD8^+^ T cells in the metastasis as compared to non-responding patients.

Since the main immunosuppressive effect of MDSC is observed in the TME, the inhibitors of their recruitment to the tumor were tested. Small molecule inhibitor of C-X-C motif chemokine receptor (CXCR) 1 and CXCR2 SX-682 was demonstrated to suppress PMN-MDSC migration and activity, and enhance the efficiency of ICI therapy in mouse oral carcinoma and Lewis lung carcinoma model [58]. In human, SX-682 has been recently applied to advanced melanoma patients alone or in combination with pembrolizumab (Table 1). This table contains ongoing clinical trials, including the combination of ICI with targeting of various immunosuppressive cells (MDSC, CAF, TAM, Treg) and tumor cells as well as with targeting of processes and molecules such as hypoxia, microbiome, neoantigens, and epigenetic mutations. In addition, we included trials combining classical ICI with targeted therapies and new immune checkpoint molecules as well.

### 4.2. Neutrophils

Exposed by high amounts of TGF-β, granulocyte-colony stimulating factor (G-CSF) and IFN-β, tumor associated neutrophils (TAN) lose their anti-tumor functions and start to support tumor progression [59]. TAN have been described to enhance tumor angiogenesis and promote metastasis [60]. High neutrophil to lymphocyte ratio (≥4) at the baseline is considered as a powerful prognostic factor associated with reduced PFS and OS in melanoma patients treated with immune checkpoint inhibitors [61,62].

### 4.3. Cancer-Associated Fibroblasts (CAF)

CAF are a major component of the tumor stroma [63]. They produce different cytokines such as TGF-β, fibroblast growth factor 2 (FGF-2) and vascular endothelial growth factor (VEGF), which lead to the tumor progression [64]. Moreover, an accumulation of CAF was described to correlate with low efficiency of anti-PD-1 therapy [65]. CAF secret fibroblast activation protein (FAP), which suppresses T cells function and recruitment [66,67]. In addition, FAP was reported to be a negative prognostic marker in the absence of immunotherapy but a positive indicative biomarker in ICI treated melanoma patients with a positive impact on PFS and OS [65]. In the murine melanoma model it was shown that stromal fibroblast matrix metalloproteinase-9 mediated surface PD-L1 cleavage, thus leading to the anti-PD-1 therapy resistance [68]. There is an ongoing trial (NCT03875079) to investigate the activity of the FAP-targeting agent RO6874281 in combination with pembrolizumab.

### 4.4. Tumor-Associated Macrophages (TAM)

TAM are known to produce interleukin (IL)-1β, cyclooxygenase-2, angiotensin, IFN-γ promoting tumorigenesis [69]. These cells can recruit regulatory T cells (Treg) and inhibit effector T cells by secreting IL-10 and expressing PD-L1 [70]. CD68^+^ TAM in tumor cell nests were described to be associated with a negative prognosis and recurrence in cutaneous melanoma [70]. Furthermore, the ratio of CD8^+^ T cells to CD68^+^ macrophages was shown to predict a disease specific survival in melanoma [71]. CD163^+^ macrophages were reported to accumulate in the TME of melanoma patients resistant to ICI therapy and to play a role in the maintenance of the immunosuppression. The depletion of CD163^+^ macrophages led to the invasion of activated T cells and inflammatory monocytes into the tumor, resulting in tumor regression [72,73].

### 4.5. Regulatory T Cells

Treg represent another important part of TME. It has been shown that the amount of forkhead box protein P3 positive (FOXP3^+^) Treg is upregulated in the peripheral blood of melanoma patients [74]. Furthermore, the frequency of circulating FOXP3^+^ Treg is associated with a poor prognosis in melanoma [75]. Tumor infiltrating Treg have been described to be a predominant cluster of the cells with high CTLA-4 expression [76]. It was found that the therapy with common anti-CTLA-4 antibodies (ipilimumab) did not deplete Treg in the tumor [77], however, Fc-engineered anti-CTLA-4 antibodies can specifically deplete FOXP3^+^ Treg and promote CD8^+^ T cell expansion, suggesting their higher clinical efficiency than the widely used non-Fc-engineered ipilimumab [76]. In another study, it was reported that the presence of Fcγ receptor-expressing macrophages within the TME is critical for the depletion of tumor-infiltrating Treg [78].

The application of NKTR-214, an engineered cytokine with biased IL-2 receptor binding, was demonstrated to selectively stimulate CD8^+^ T cells and to deplete Treg in patients with advanced or metastatic solid tumors [79].

## 5. Role of Microbiome in the ICI Therapy of Melanoma

It has recently been clearly demonstrated that the microbiome could influence the ICI therapy in melanoma patients [80]. Although oral microbiome showed no effect on the response to cancer immunotherapy, an enrichment of *Clostridiales*, *Ruminococcaceae*, and *Faecalibacterium* in the gut was associated with response, while an enrichment of *Bacteroidales* was observed in non-responders and associated with increased risk of relapse [80]. The same study demonstrated that a favorable gut microbiome composition at the baseline was associated with increased CD8^+^ T cell infiltration and anti-tumor immune responses. Furthermore, the fecal transplantation from melanoma patients responding to ICI to germ-free mice led to a better response to anti-PD-1 therapy as compared to mice, receiving gut transplants from non-responding patients [80]. Another study demonstrated that the presence of *Bifidobacterium longum*, *Collinsella aerofaciens*, and *Enterococcus faecium* was associated with a better prognosis in melanoma patients [81]. Moreover, the anti-cancer immunity was described to be affected by the alteration in the metabolism of specific bacterial species but not by their presence [82]. There are several ongoing clinical trials dealing with the gut microbiota transplantation in melanoma patients (Table 1).

## 6. Predicting the Response to the ICI Therapy

Since the response rates to ICI treatment are still restricted [26,27,28,29,83], the identification of response-biomarkers before or shortly after the therapy initiation is one of the biggest challenges in the immuno-oncology. Current approaches to predict response to ICI in melanoma are based on the radiology, tumor biopsy and liquid biopsy [84,85].

Radiological imaging (body computer tomography (CT) scan, head magnetic resonance imaging (MRI)) is used to assess the response to ICI treatment in melanoma patients and is routinely performed three months after the start of treatment. Prediction of response in the earlier time points is possible by using ^18^F-FDG PET/CT, where response criteria were developed using the scans made at 21 to 28 days after the start of treatment [86]. This approach was also shown to be beneficial in long-term response prediction and guidance of ICI withdrawal [87,88,89].

As a part of PD-1/PD-L1 axis, amount of PD-L1 expression on tumor cells was thought to be a distinct predictive marker for therapy response. Although PD-L1 overexpressing tumors showed an association with the higher response to ICI, durable responses could be also observed in PD-L1 negative tumors [90,91]. Therefore, complementary approaches are needed to improve the prognostic value of tumor PD-L1, including a dynamic monitoring of PD-L1 expression or PD-L1 RNA sequencing [92,93].

Further interest attracts the measurement of PD-L1 (soluble and expressed in extracellular vesicles, EV) in liquid biopsies. Soluble PD-L1 is a splice variant without a transmembrane domain capable to directly inhibit T cell proliferation and IFN-γ production [94]. Elevated basal levels of soluble PD-L1 in the plasma of melanoma patients was associated with progressive disease [95]. Furthermore, the measurement of PD-L1 in EV could help to predict the response to ICI, demonstrating an advantage of the detection in EV over tumor biopsies [96]. Melanoma patients responding to pembrolizumab could be distinguished from non-responders by increased levels of EV PD-L1 at 3 to 6 weeks after the start of therapy [97]. In another study, it was shown that exosomal PD-L1 mRNA levels decreased during nivolumab or pembrolizumab treatment of melanoma patients with complete or partial response, while in patients with progressive disease EV PD-L1 expression was increased [98].

Besides PD-L1, soluble CD163 and macrophage-related chemokines (e.g., C-X-C motif chemokine ligand (CXCL) 5, 10) were reported to predict efficacy of ICI [85]. Decreased serum levels of IL-8 at 2 to 4 weeks after the start of ICI treatment were associated with the response in patients even with the initial pseudoprogression [99]. Induction of CXCR3 ligands in murine melanoma model was described to increase the response to the therapy with anti-PD1 antibodies, and elevated CXCR3 levels were observed in plasma of responding melanoma patients [100].

Another predictive marker could be the amount of tumor-infiltrating T cells. It has been shown that T cells dominated among other immune cells, accumulated in human melanoma metastatic tissue [101]. Strong pre-existing T cell infiltration, IFN-γ–related gene expression signatures in the tumor and high serum level of IFN-γ were reported to be associated with a good clinical prognosis and to predict the response to anti-PD-1 therapy in melanoma patients [101,102,103,104,105]. It was reported that 98% of PD-L1^+^ tumors were associated with high TIL numbers and the PD-L1^+^ melanoma cells were localized adjacent to TILs [106].

## 7. Increasing Effectiveness of ICI Therapy

In order to enhance the beneficial therapeutic effect of ICI, this treatment was combined with other anti-tumor therapies. Since radiation therapy (RT) is used in melanoma patients and can induce antigen release from tumors, its combination with immunotherapy was applied, leading to the T cell activation and improvement of OS without increasing the number of adverse events [107,108]. In a retrospective study with 208 melanoma patients with brain metastasis treated with anti-PD-1 antibodies and RT, the survival rates at 6 and 12 months after the start of treatment were 77% and 70%, respectively [109]. There are numerous ongoing trials investigating the combination of immuno- and radiation therapy in metastatic melanoma patients (Table 1).

Another promising approach to increase the efficiency of ICI is to combine it with metformin, a drug for type II diabetes. Metformin was shown to induce not only cell cycle arrest in melanoma cells, leading to their autophagy and apoptosis, but also to affect the TME [110]. It is known that metformin activates AMP-activated protein kinase a (AMPKa) in mitochondria, which lead to the downregulation of HIF-1α expression, resulting in reduced intratumoral hypoxia. Metformin was also reported to promote T cell activity in the combination with ICI, leading to B16 melanoma rejection in mice [111]. In a clinical trial, it was shown that the combination of ICI and metformin increased objective response rate (ORR), disease control rate (DCR), PFS and OS in comparison with the group treated with ICI alone [112]. However, due to a small patient cohort, these changes were not statistically significant.

Interestingly, the reduction of tumor hypoxia could be achieved by a physical exercise as well. In B16F10 mouse melanoma model, voluntary wheel running resulted in the epinephrine-dependent, IL-6-sensitive NK cell activation and increased migration of NK and T cells into the tumor [113]. In addition, a physical activity prior to tumor cells inoculation led to a strong reduction of primary tumor growth and numbers of lung metastasis in those mice. Other study demonstrated that the growth of B16F10 melanoma in mice on high-fat diet was accelerated as compared to mice receiving a balanced diet [114]. Importantly, this growth increase was significantly reduced by continuous physical exercise that was associated with the lymphocyte proliferation [114]. In melanoma patients, exercises undertaken before diagnosis were not significantly correlated with a reduction in cancer-related or overall mortality [115]. However, in patients with unresectable stage III or IV melanoma undergoing immunotherapy, the reduction of fatigue was shown to be the main positive impact of physical activity [116]. The ongoing combinational trial is represented in the Table 1.

Targeted therapies (BRAF and MEK-inhibitors) are known to be effective in patients with BRAF-V600 mutation and achieve rapid response with a high response rate [117]. The median maintenance of response to this therapy is approximately one year because of the development of acquired resistance [118], while ICI have been described to induce durable response. It was reported that 33% of melanoma patients achieved complete response when treated with the combination of dabrafenib and trametinib with spartalizumab (anti-PD-L1-antibody); the 1-year OS was 86%; however, the number of grade ≥3 adverse events was 75% [119]. In another study, dabrafenib and trametinib were combined with pembrolizumab (triple therapy) or placebo (double therapy) [120]. The median duration of response in tripled therapy group was 18.7 months and 12.5 months in double therapy group. PFS was 16.0 months in triple and 10.3 months in double therapy. In a smaller patient’s cohort, an objective response was achieved in 73% of patients, and 40% maintained the response at a median follow-up of 27.0 months [121]. 73% of patients from the same cohort developed grade 3 and 4 adverse events. Another trial, investigating the combination of atezolizumab (anti-PD-L1-anibody), cobimetinib and vemurafenib showed similar results with an objective response rate of 71.8% and median duration of response of 17.4 months; 39.4% of patients maintained response for 29.9 months of follow-up [122]. These data suggest that this combination therapy can increase the maintenance of the response, but the high grades of adverse events need to be taken into account. Ongoing trials to the triple combination are shown in Table 1.

ICI could also be combined with the oncolytic virus talimogen laherparepvec (T-VEC) that was approved for melanoma immunotherapy. T-VEC is a genetically modified virus, which replicates in tumor cells causing cancer cell lysis [123]. It has been reported that the intratumoral T-VEC injection in combination with pembrolizumab led to increased CD8^+^ T cells infiltration associated with the ORR rate of 62% and the CR in 33% of patients [124].

Combination of all-trans retinoic acid (ATRA) with ipilimumab was reported to decrease frequency of circulating MDSC as well as the expression of PD-L1, IL-10, and indoleamine 2,3-dioxygenase by MDSC, whereas in the ipilimumab monotherapy group the MDSC frequency increased during the treatment [125]. Furthermore, patients receiving combinational treatment tend to have an increased activated CD107a^+^ IFN-γ^+^CD8^+^ T cell numbers compared to the patients treated with ipilimumab alone.

Combination of NKTR-214 and Nivolumab was shown to achieve response rates of 53%, which correlated with high IFN-γ levels [126]. Furthermore, the accumulation of IFN-γ and CD8^+^ TIL in tumor tissue had been seen in favorable as well as in unfavorable tumor microenvironment. The ongoing trials investigating the combination of NKTR-214 with ICI in metastatic melanoma patients are listed in Table 1.

It was demonstrated that epigenetic modulation induced by the histone deacetylase inhibitor entinostat (MS-275) could enhance the antigen presentation in tumor cells and inhibit immunosuppressive activity of MDSC and Treg [127,128]. After combining entinostat with the anti-PD-1 antibodies, 19 % of non-responding to anti-PD-1 therapy melanoma patients, achieved objective response [129]. These data represent a new approach to overcome resistance using epigenetics. Other ongoing trials using this combination are listed in Table 1.

A new approach of targeting different TME components using nanoparticles has been recently proposed [130]. In melanoma mouse models, nanoparticles were shown to potentiate the efficiency of PD-1 blockade [131,132,133], to reduce the tumor volume and to prolong mouse survival [134].

## 8. Other ICI in Malignant Melanoma

In addition to PD-L1 and CTLA-4, several other immune checkpoint molecules have been investigated during the last decade. Among them are lymphocyte activation gene-3 (LAG-3), T-cell immunoglobulin- and mucin domain- containing molecule 3 (TIM-3) and T cell immunoreceptor with Ig and ITIM domains (TIGIT). All these molecules were reported to be highly expressed on immune cells in the TME, especially on TILs and Treg, which makes them a promising target for cancer immunotherapy [135].

LAG-3 is expressed on activated CD4^+^ and CD8^+^ T cells, Treg, B and NK cells as well as DC [136]. It interacts with MHCII molecules on APC or with Galectin-3 and liver sinusoidal endothelial cell lectin (LSECtin) on cancer cells, leading to the inhibition of CD4^+^ and CD8^+^T cell proliferation and decreased cytokine secretion [137]. Such inhibition of T cell function was found to be associated with the promotion of tumor growth and tumor escape [138,139]. LAG-3 blocking could be achieved by LAG-3-Ig fusion protein or LAG-3 targeting antibody (relatlimab). The treatment of melanoma patients with relatlimab resulted in the ORR of 16% and DCR of 45% [140]. Interestingly, only 9% of patients had grade 3 or 4 adverse events that was comparable to the therapy with nivolumab.

TIM-3 is expressed on CD4^+^ and CD8^+^ T cells, Treg, B cells, NK cells, DC, mast cells and macrophages. Under physiological conditions, it serves as a negative regulator of Th1 response and Th1 related production of TNF and IFN-γ; therefore, its blockade could lead to autoimmune disease [141]. Interaction of TIM-3 with Galectin-9 expressed on tumor cells was reported to result in CD8 TIL apoptosis in colon cancer [142]. In melanoma high expression of TIM-3 was associated with CD8 T cell exhaustion [143].

TIGIT was reported to be involved in the inhibition of CD8^+^T cells and modulation of DC activity, resulting in the upregulation of IL-10 and downregulation of IL-12 production [144,145]. Moreover, TIGIT was demonstrated to play a crucial role in the maturation of naïve T cells to Foxp3^+^ Treg [146]. TIGIT^+^ Tregs showed higher immunosuppressive potential than their TIGIT^-^ counterparts [147]. In malignant melanoma, the co-expression of PD-L1, LAG-3, TIM-3 and TIGIT was demonstrated to induce CD8^+^ TILs with most exhausted phenotype [125,126]. Double blockage of PD-1 and TIGIT in melanoma led to an increased proliferation and cytokine production of CD8^+^ TIL and was considered to be a promising approach in immunotherapy [148]. The ongoing clinical trials evaluating the efficiency of LAG-3, TIM-3 and TIGIT blockade are shown in Table 1.

## 9. Conclusions

Despite of melanoma immunogenicity, this tumor develops immune escape mechanisms that stimulate a fast melanoma progression. Such mechanisms include impaired antigen presentation by tumor cells, accumulation of dysfunctional effector T cells and generation of the immunosuppressive TME represented by MDSC, TAN, CAF, TAM, and Treg. Therefore, numerous approaches were developed to reinvigorate the anti-tumor immune response. Recently approved immunotherapies with ICI (anti-PD-1, anti-PD-L1 and anti-CTLA-4 antibodies) have revolutionized the treatment of melanoma. This treatment significantly increased the survival of melanoma patients and provided a durable control of the disease [26,27,28]. However, the response rates to ICI are still restricted. Thus, further efforts should be undertaken to maximize the efficacy of ICI treatment. This aim could be achieved by improving the selection of patients who might benefit from the ICI therapy, by applying early radiological findings and by measuring predictive markers from tumor and liquid biopsies. Furthermore, the combination of different ICI (such as ipilimumab and nivolumab), their combination with targeting of the immunosuppressive TME or with other anti-cancer therapies could significantly improve the efficacy of tumor immunotherapy. Furthermore, targeting of other immune checkpoints (such as LAG-3, TIM-3, TIGIT) and its combination with approved ICI are currently under investigation (Table 1). Approved ICI, their targets, and targets for combined treatments are summarized in the Figure 1.

## Figures and Tables

**Figure 1 ijms-21-02367-f001:**
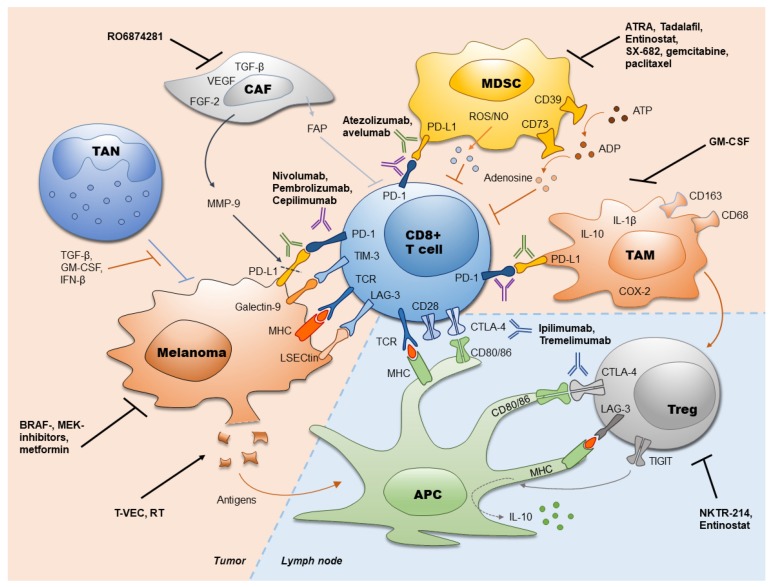
Immune checkpoint inhibitors in melanoma and their combination with other therapies. Currently used antibodies against PD-1 (atezolizumab, avelumab), PD-L1 (nivolumab, pembrolizumab, cepilimumab) and CTLA-4 (ipilimumab, tremelimumab) as well as strategies to increase the efficiency of immune checkpoint inhibitors (ICI) are presented. ADP: adenosine diphosphate; APC: antigen presenting cell; ATP: adenosine triphosphate; ATRA: all-trans retinoic acid; CAF: cancer-associated fibroblasts; COX-2: cyclooxygenase-2; CTLA-4: cytotoxic T lymphocyte-associated protein-4; FAP: fibroblast activation protein; FGF-2: fibroblast growth factor 2; GM-CSF: granulocyte-macrophage colony stimulating factor; IFN-β: interferon-β; IL: interleukin; LAG-3: lymphocyte activation gene-3; LSECtin: liver sinusoidal endothelial cell lectin; MDSC: myeloid-derived suppressor cells; MHC: major histocompatibility complex; MMP-9: matrix metallopeptidase 9; NO: nitric oxide; PD-1: programmed cell death protein 1; PD-L1: programmed cell death ligand 1; ROS: reactive oxygen species; RT: radiation therapy; TAM: tumor-associated macrophages; TAN: tumor associated neutrophils; TCR: T-cell receptor; TGF-β: transforming growth factor-β; TIGIT: T cell immunoreceptor with Ig and ITIM domains; TIM-3: T-cell immunoglobulin- and mucin domain- containing molecule 3; Treg; regulatory T cells; T-VEC: talimogen laherparepvec; VEGF: vascular endothelial growth factor.

**Table 1 ijms-21-02367-t001:** Ongoing combinatorial clinical trials.

Targets	Trial Number	Intervention	Disease	Trial Phase
MDSC	NCT03200847	ATRA (Vesanoid) + pembrolizumab	Advanced melanoma	I, II
	NCT02403778	ATRA + ipilimumab	Advanced melanoma	II
	NCT03161431	SX-682 alone or in combination with pembrolizumab	Melanoma (III, IV)	I
	NCT02259231	RTA 408 (Omaveloxolone) + nivolumab or ipilimumab	Unrespectable or metastatic melanoma	Ib, II
CAF	NCT03875079	RO6874281 + pembrolizumab	Metastatic melanoma	Ib
TAM	NCT01363206	GM-CSF (Leukine, Sargramostim) + ipilimumab	Unresectable metastatic melanoma	II
Treg	NCT02203604	Aldesleukin (IL-2) + ipilimumab	Metastatic melanoma (IIIA–IV)	II
	NCT02983045	NKTR-214 (PEGylated IL-2) + nivolumab with or without ipilimumab	Advanced malignancies, including melanoma	I, II
	NCT03548467	NKTR-214 after prior anti-PD-1 therapy	Advanced malignancies, including melanoma	I, II
	NCT03635983	NKTR-214 + nivolumab or nivolumab alone	Untreated, inoperable or metastatic melanoma	III
	NCT03138889	NKTR-214 + pembrolizumab	Advanced malignancies, including melanoma	I, II
	NCT03435640	Intratumoral NKTR-262 + systemic NKTR-214 with or without nivolumab	Melanoma and other cancer types	I, II
	NCT03635983	NKTR-214 + nivolumab or nivolumab alone	Untreated, inoperable or metastatic melanoma	III
Microbiome	NCT03341143	Fecal microbiota transplant (FMT) + pembrolizumab	Advanced melanoma patients, non-responders	II
	NCT03817125	Vancomycin or placebo pretreatment + nivolumab + SER-401 or placebo	Unresectable or metastatic melanoma	Ib
	NCT03772899	FMT for a healthy donor a week before approved melanoma treatment (pembrolizumab/nivolumab)	Advanced melanoma	I
	NCT03643289	Comparison of gut microbiome before and during anti-PD-1 therapy (till week 9)	Advanced melanoma stage IV	Observational
Hypoxia	NCT03311308	Metformin + pembrolizumab or pembrolizumab alone	Advanced, unresectable melanoma stage III or IV	I
	NCT03171064	Exercise + nivolumab or pembrolizumab	Metastatic melanoma	II
Tumor cells	NCT02799901	Hypofractionated radiation therapy (RT) (27 Gy over 3 fractions) + nivolumab	Advanced melanoma	II
	NCT03693014	Hypofractionated RT + Ipilimumab, Nivolumab or Pembrolizumab, continued according to the standard schedule	Metastatic cancer, including melanoma	II
	NCT02406183	Ipilimumab + RT	Metastatic melanoma	I
	NCT04042506	Nivolumab + RT	Metastatic melanoma	II
	NCT04017897	Anti-PD1 (pembrolizumab or nivolumab) + RT	Unresectable, naive metastatic melanoma (IIIB to IVM1c)	II
	NCT01449279	Ipilimumab + RT	Metastatic melanoma	II
	NCT01689974	Ipilimumab + RT or ipilimumab alone	Metastatic melanoma	II
	NCT01769222	Ipilimumab + RT or ipilimumab alone	Recurrent malignancies, including melanoma	I, II
	NCT02659540	Nivolumab + ipilimumab in combination with conventional or hypofractionated RT	Unresectable melanoma stage IV	I
	NCT02263508	Pembrolizumab + T-VEC or placebo	Stage IIIB-IVM1c melanoma	III
	NCT04068181	Pembrolizumab + T-VEC after progression on anti-PD-1 therapy	Stage IIIB-IVM1d melanoma	II
	NCT01740297	Ipilimumab + T-VEC or ipilimumab alone	Stage IIIB–IV metastatic melanoma	I, II
	NCT02965716	Pembrolizumab + T-VEC	Stage IIIB–IV metastatic melanoma	II
	NCT03842943	Neoadjuvant pembrolizumab + T-VEC	Resectable stage 3 melanoma	II
Tumor mutations	NCT02902042	Encorafenib + binimetinib + pembrolizumab	Metastatic BRAF V600 mutant melanoma	I, II
	NCT02910700	Nivolumab + trametinib with or without dabrafenib	BRAF-mutated or wild type metastatic stage III-IV melanoma	II
	NCT02908672	Cobimetinib + vemurafenib with atezolizumab or placebo	Metastatic BRAF V600 mutant melanoma	III
	NCT02303951	Vemurafenib + cobimetinib + atezolizumab	BRAF V600 mutant stage IIIC-IV melanoma	II
	NCT01767454	Dabrafenib + ipilimumab or dabrafenib + trametinib + ipilimumab	Metastatic or unresectable BRAF V600 mutant melanoma	I
Epigenetic modifications	NCT03765229	Entinostat + pembrolizumab	Stage III–IV metastatic melanoma	II
	NCT02437136	Entinostat + pembrolizumab	Advanced malignancies, including melanoma	Ib, II
Neoantigens	NCT03929029	NeoVax + Montanide^®^ with nivolumab + ipilimumab	Advanced melanoma	Ib
	NCT02385669	Peptide Vaccine + Ipilimumab	Stage IIA–IV melanoma (advanced, adjuvant, neoadjuvant)	I, II
	NCT03047928	PD-L1/IDO peptide vaccine + nivolumab	Metastatic melanoma	I, II
	NCT03633110	GEN-009 Adjuvant Vaccine + pembrolizumab or nivolumab	Solid tumors, including melanoma	I, II
	NCT04072900	Personalized neoantgen peptide vaccine + anti-PD-1 + rhGM-CSF + Imiquimod 5% Topical Cream	Metastatic melanoma	I
	NCT04091750	Nivolumab + ipilimumab + cabozantinib followed by nivolumab + cabozantinib	Advanced melanoma	II
Other immune checkpoint molecules	NCT02676869	IMP321 + pembrolizumab	Stage III–IV advanced melanoma	I
	NCT02519322	Nivolumab + relatimab or + ipilimumab or alone before surgery	Stage IIIb–IV advanced melanoma	II
	NCT03743766	Relatimab + nivolumab or each drug alone followed by relatimab + nivolumab in all subjects	Unresectable or metastatic melanoma	II
	NCT03470922	Relatimab + nivolumab or nivolumab alone	Unresectable or metastatic melanoma	II, III
	NCT03652077	INCAGN02390 antibody against TIM-3 alone	Advanced malignancies, including melanoma	I
	NCT04139902	Neoadjuvant therapy with PD-1 inhibitor dostarlimab (TSR-042) or dostarlimab (TSR-042) + TSR-022 (TIM-3 inhibitor)	Stage IIIB–IV advanced melanoma	II
	NCT03708328	RO7121661, bispecific anti-PD-1 and anti-TIM-3 antibody	Advanced malignancies, including melanoma	I
	NCT02817633	TSR-022 (anti-TIM-3) alone or + TSR-042 (anti-PD-1) or triple combination of TSR-022 (anti-TIM-3), TSR-042 (anti-PD-1) and TSR-033 (anti-LAG3)	Advanced malignancies, including melanoma	I
	NCT03628677	AB154 (anti-TIGIT) alone or + AB122 (anti-PD-1)	Advanced malignancies, including melanoma	I
	NCT03119428	OMP-313M32 (anti-TIGIT) alone or + nivolumab	Advanced malignancies, including melanoma	I

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
