# Peer review of "Modern Aspects of Immunotherapy with Checkpoint Inhibitors in Melanoma"

_ijms, 2020, doi:10.3390/ijms21072367_

Round 1

Reviewer 1 Report

The manuscript by Petrova et al., “Modern aspects of immunotherapy with checkpoint inhibitors in melanoma” is a well written and organized, extensive and timely manuscript giving an up-date on the current status of immune check point therapy in melanoma.

I suggest the authors to consider adding a paragraph or two discussing the predictive value of T cells in tumors (to part 5. Predicting the response to the ICI therapy). Although there is not full consensus, the majority of papers have shown that high numbers of T cells in tumors is associated with response, and similarly concerning IFN-g signaling.

Along those lines, I also suggest adding a paragraph or two to part 7. Increasing effectiveness of ICI therapy. I think it would be worthwhile mentioning that exercise is being increasingly studied as a potential combination partner for ICI therapy, due to the effect exercise related adrenalin has on mobilization of key effector cells of the immune system; T and NK cells. To this end, several reviews has been published recently and the numbers of trials studying the potential therapeutic effect of exercise published on www.clinicaltrial.gov is rapidly increasing

Author Response

Reviewer #1:

Comment 1: “Consider adding a paragraph or two discussing the predictive value of T cells in tumors (to part 5. Predicting the response to the ICI therapy). Although there is not full consensus, the majority of papers have shown that high numbers of T cells in tumors is associated with response, and similarly concerning IFN-g signaling.”

Response 1: We thank the Reviewer for the comment. We included the requested changes in the revised version (p. 8, lines 234-240).

Comment 2: “I also suggest adding a paragraph or two to part 7. Increasing effectiveness of ICI therapy. I think it would be worthwhile mentioning that exercise is being increasingly studied as a potential combination partner for ICI therapy, due to the effect exercise related adrenalin has on mobilization of key effector cells of the immune system; T and NK cells. To this end, several reviews has been published recently and the numbers of trials studying the potential therapeutic effect of exercise published on www.clinicaltrial.gov is rapidly increasing.”

Response 2: We thank the reviewer for this important suggestion. We included a paragraph on the impact of exercises as a potential combination partner for ICI therapy in the chapter 7 (p. 8, lines 260-271).

We thank Reviewers for the constructive criticism, which led to an improvement of the review.

Sincerely yours,

Viktor Umansky, Ph.D.  

Reviewer 2 Report

The review is well structured, contains the most relevant references and is well written.

Authors focus on the role of immune checkpoint inhibitors in human melanoma and their combination with other therapeutic options.

Here are some points which need clarification or correction:

Major points:

1) The table represents an important aspect of this manuscript. The generation of this table should be explained. By what kind of criteria where the important trials identified. Additionally the purpose of the table should be mentioned.

2) The manuscript would benefit when chapter 5. and 6. are swapped (more logical).

3) PD-L1 and PD-1 are currently the most important targets for ICI. In the Figure PD-L1 expression is shown on melanoma and MDSCs. Macrophages and other APC are also PD-L1 positive, which is important when treating patients. Do the authors want to comment on this issue?

4) The conclusion is very minimalistic. A reappraisal and discussion of the former text would be appropriate.

Minor points:

15: “in past decade” -> in the past decade

20: “present review” -> this review

24: “approaches by targeting of other” -> approaches targeting other

30-31: wording of the sentence could be better (“has been shown … as described in the

concept of”)

35: “by immune system” -> by the immune system

49: wording/grammar could be better (“Another mechanism of beneficial effect”)

53: “In this review, we will focus” -> This review will focus

53: “check point” -> checkpoint

67: “5-years” -> 5-year

71: in row 67 “5-year” written with a hyphen, now “1 year” written without

73: “such combination” -> such combinations

81: “genes, regulating” -> genes regulating

108: space is missing between the last word of the sentence and the references

115: “could be also achieved” -> could be achieved

119: “data, showing” -> data showing

128: PMN misses an explanation/the whole word

132: the heading line of the table could be repeated on every page

132: the table would be better at the end of chapter 4, not in the middle of it

134: abbreviations for “granulocyte-colony stimulating factor” and “interferon-beta” would

be good, as for all other factors abbreviations are stated

134-136: wording/grammar of the sentence could be better

137: “angiogenesis in the tumor” -> tumor angiogenesis

138: “considered as a” -> considered a

141: “produced” -> produce

142: abbreviations for “fibroblast growth factor 2” and “vascular endothelial growth factor”

would be good, as for all other factors abbreviations are stated

143: a better connection between the rest of the text and the statement “expressing smooth

muscle actin” would be good

162: “forkhead box protein P3 (FOXP3+) Treg” -> forkhead box protein P3 positive (FOXP3+)

Treg

167: wording/grammar could be better (“suggesting their better clinical efficiency than

ipilimumab”

196: “Melanoma patients, responding” -> Melanoma patients responding

203: CXCL5 and CXCL10 are stated and in row 205 CXCL3 is mentioned -> is there a

difference between the different CXCLs?

215-217 and 219-221: What is the different between the two statements?

242-243: wording/grammar could be better (“…that could be explained…”

248: what is “CR”? Before only “DCR” is mentioned, is it the same?

249: before the survival percentage was always stated without a decimal

257: “response rate” -> response rate of

258: “response” -> response of

263: “is genetically” -> is a genetically

279: why “Entinostat” capitalized?

284-285: wording/grammar could be better (“It has been recently proposed a new

approach…”)

299: “LAG-3blocking” is missing a space

343: the abbreviation “RT” is stated two times

Author Response

Reviewer #2:

Comment 1: “The table represents an important aspect of this manuscript. The generation of this table should be explained. By what kind of criteria where the important trials identified. Additionally, the purpose of the table should be mentioned.”

Response 1: As suggested by the Reviewer, we explained how the table was generated and described the purpose of the table in the revised version (p. 3, line 131-135).

Comment 2: “The manuscript would benefit when chapter 5. and 6. are swapped (more logical).”

Response 2: We agree with the Reviewer and swapped the chapter 5 and 6.

Comment 3: “PD-L1 and PD-1 are currently the most important targets for ICI. In the Figure PD-L1 expression is shown on melanoma and MDSCs. Macrophages and other APC are also PD-L1 positive, which is important when treating patients. Do the authors want to comment on this issue?”

Response 3: We thank the Reviewer for this valuable comment. We corrected the Figure 1 by including PD-L1-expressing macrophage (TAM) as well as tumor-associated neutrophil (TAN) and a cancer-associated fibroblast (CAF). We mentioned also intext that TAM can express PD-L1 (p. 4, lane 156-157).

Comment 4: “The conclusion is very minimalistic. A reappraisal and discussion of the former text would be appropriate.”

Response 4: As suggested by the Reviewer, we revised the conclusion (p. 10, lines 346-361).

Minor comments:

Comment 1: 15: “in past decade” -> in the past decade

Response 1: We thank the Reviewer for this comment and revised as requested.

Comment 2: 20: “present review” -> this review

Response 2: We thank the Reviewer for this comment and revised as requested.

Comment 3: 24: “approaches by targeting of other” -> approaches targeting other

Response 3: We thank the Reviewer for this comment and changed as requested.

Comment 4: 30-31: wording of the sentence could be better (“has been shown … as described in the concept of”)

Response 4: According to the Reviewer’s suggestion, we corrected the wording of the sentence.

Comment 5: 35: “by immune system” -> by the immune system

Response 5: We corrected as suggested.

Comment 6: 49: wording/grammar could be better (“Another mechanism of beneficial effect”)

Response 6: According to the Reviewer’s suggestion, we corrected the wording.

Comment 7: 53: “In this review, we will focus” -> This review will focus

Response 7: We thank the Reviewer for this comment and changed as proposed.

Comment 8: 53: “check point” -> checkpoint

Response 8: We thank the Reviewer for this comment and changed as suggested.

Comment 9: 67: “5-years” -> 5-year

Response 9: We corrected as proposed.

Comment 10: in row 67 “5-year” written with a hyphen, now “1 year” written without

Response 10: We wrote “1 year” with a hyphen.

Comment 11: 73: “such combination” -> such combinations

Response 11: We thank the Reviewer for this comment and corrected as suggested.

Comment 12: 81: “genes, regulating” -> genes regulating

Response 12: We changed as suggested.

Comment 13: 108: space is missing between the last word of the sentence and the references

Response 13: We thank the Reviewer for this comment and corrected as suggested.

Comment 14: 115: “could be also achieved” -> could be achieved

Response 14: We revised as suggested.

Comment 15: “data, showing” -> data showing

Response 15: We revised as suggested.

Comment 16: 128: PMN misses an explanation/the whole word

Response 16: We included the explanation of PMN in the lane 102, where it is mentioned for the first time.

Comment 17: 132: the heading line of the table could be repeated on every page Response 17: We thank the Reviewer for this comment and changed as suggested.

Comment 18: 132: the table would be better at the end of chapter 4, not in the middle of it

Response 18: We thank the Reviewer for this comment and changed as suggested.

Comment 19: 134: abbreviations for “granulocyte-colony stimulating factor” and “interferon-beta” would be good, as for all other factors abbreviations are stated Response 19: We thank the Reviewer for this comment and provided the requested abbreviations.

Comment 20: 134-136: wording/grammar of the sentence could be better

Response 20: We thank the Reviewer for this comment and improved the wording (lane 137-139).

Comment 21: 137: “angiogenesis in the tumor” -> tumor angiogenesis

Response 21: We revised as requested.

Comment 22: 138: “considered as a” -> considered a

Response 22: We revised as requested.

Comment 23: 141: “produced” -> produce

Response 23: We corrected as requested.

Comment 24: abbreviations for “fibroblast growth factor 2” and “vascular endothelial growth factor” would be good, as for all other factors abbreviations are stated

Response 24: We provided the abbreviations as requested.

Comment 25: 143: a better connection between the rest of the text and the statement “expressing smooth muscle actin” would be good

Response 25: We corrected the sentence removing “expressing smooth muscle actin” (lane 146).

Comment 26: 162: “forkhead box protein P3 (FOXP3+) Treg” -> forkhead box protein P3 positive (FOXP3+) Treg

Response 26: We changed as requested.

Comment 27: 167: wording/grammar could be better (“suggesting their better clinical efficiency than ipilimumab”)

Response 27: We corrected the sentence writing “suggesting their higher clinical efficiency than widely used non-Fc-engineered ipilimumab” (now lane 172)

Comment 28: 196: “Melanoma patients, responding” -> Melanoma patients responding

Response 28: We corrected as requested (now lane 220).

Comment 29: 167: wording/grammar could be better (“suggesting their better clinical efficiency than ipilimumab”)

Response 29: We corrected the sentence writing “suggesting their higher clinical efficiency than widely used non-Fc-engineered ipilimumab” (now lane 172)

Comment 30: 203: CXCL5 and CXCL10 are stated and in row 205 CXCL3 is mentioned -> is there a difference between the different CXCLs?

Response 30: We thank the Reviewer for this question. In the manuscript, we refer to the CXCR3 ligands but not to the CXCL3.

Comment 31: 215-217 and 219-221: What is the different between the two statements?

Response 31: We thank the Reviewer for this question. In the revised version, we removed the statement “These authors have also shown that the transplantation of gut microbiota from melanoma patients responding to ICI to germ-free mice improved the efficiency of the anti-PD-1 therapy in mice” (lane 219-221 in the original version).

Comment 32: 242-243: wording/grammar could be better (“…that could be explained…”

Response 32: We thank the Reviewer for this comment. We corrected this sentence: “However, due to a small patient cohort, these changes were not statistically significant” (now lane 254-255).

Comment 33: 248: what is “CR”? Before only “DCR” is mentioned, is it the same?

Response 33: CR means complete response. We wrote now “complete response” instead of CR (now lane 272).

Comment 34: 249: before the survival percentage was always stated without a decimal

Response 34: We thank the Reviewer for this comment. We wrote 86% without decimal in the revised version (now lane 273).

Comment 35: 257: “response rate” -> response rate of

Response 35: We corrected as suggested (now lane 281).

Comment 36: 258: “response” -> response of

Response 36: We corrected as suggested (now lane 282).

Comment 37: 263: “is genetically” -> is a genetically

Response 37: We corrected as suggested (now lane 287).

Comment 38: 279: why “Entinostat” capitalized?

Response 38: We changed to “entinostat” in the revised version (now lane 303).

Comment 39: 284-285: wording/grammar could be better (“It has been recently proposed a new approach…”)

Response 39: We changed the wording: “A new approach of targeting different TME components using nanoparticles has been recently proposed” (now lanes 308-309).

Comment 40: 299: “LAG-3blocking” is missing a space

Response 40: We included the space (now lane 322).

Comment 41: 343: the abbreviation “RT” is stated two times

Response 42: We thank the Reviewer for this comment. We mentioned the abbreviation “RT” only once (now lane 384).

We thank Reviewers for the constructive criticism, which led to an improvement of the review.

Sincerely yours,

Viktor Umansky, Ph.D.